# Novel Prediction Framework for Path Delay Variation Based on Learning Method

**Jingjing Guo****, Peng Cao**, **Zhaohao Sun, Bingqian Xu, Zhiyuan Liu and Jun Yang** *

National ASIC System Engineering Center, Southeast University, Nanjing 210096, China; guojingjing@seu.edu.cn (J.G.); caopeng@seu.edu.cn (P.C.); sunzhaohao@seu.edu.cn (Z.S.); 220181352@seu.edu.cn (B.X.); zyliuasic@seu.edu.cn (Z.L.)

* Correspondence: dragon@seu.edu.cn; Tel.: +86-025-8379-3265

**Abstract:** Path delay variation becomes a serious concern in advanced technology, especially for multi-corner conditions. Plenty of timing analysis methods have been proposed to solve the issue of path delay variation, but they mainly focus on every single corner and are based on a characterized timing library, which neglects the correlation among multiple corners, resulting in a high characterization effort for all required corners. Here, a novel prediction framework is proposed for path delay variation by employing a learning-based method using back propagation (BP) regression. It can be used to solve the issue of path delay variation prediction under a single corner. Moreover, for multi-corner conditions, the proposed framework can be further expanded to predict corners that are not included in the training set. Experimental results show that the proposed model outperforms the traditional Advanced On-Chip Variation (AOCV) method with 1.4X improvement for the prediction of path delay variation for single corners. Additionally, while predicting new corners, the maximum error is 4.59%, which is less than current state-of-the-art works.

**Keywords:** path delay variation; learning-based; multi-corner conditions

## 1. Introduction

Variation is a significant and expensive problem. Accompanying the development of integrated circuits, the feature size of technology is getting smaller and smaller and will continue shrinking in the future. However, the size of atoms does not change. Regarding advanced technology, the oxide layer of the gate is just a few atoms thick, so even a single atom out of place can change device and cell performance considerably [1]. As the voltage decreases, the dependence of the transistor current on threshold (Vth), supply voltage (Vdd) and temperature gradually changes from linear to exponential [2–6]. The gate delay variation increases by five times in near-threshold voltage [7] and two times across the entire temperature [8]. Moreover, considering modern technology, due to the complexity and variability of semiconductor processes, delay variation is no longer a simple relationship of the process, voltage, and temperature (PVT), so sensitivity of delay variation varies differently [9,10]. Therefore, the worst corner of different designs may be different and cannot be easily identified. Every corner needs to be analyzed, and corner numbers have increased by 1000 at 32 nm [11–13]. Besides, the characterizations and simulations of each corner are very long, so much time and effort are needed to verify circuit functionality in all kinds of corners. Also, low voltage has a dramatic influence on path delay variation because, at low voltage, delay variation does not obey Gaussian distribution but non-Gaussian distribution [14–20], which makes the path delay variation more difficult to predict.

Considering variation, various techniques are proposed, such as post-silicon tuning techniques [21,22], but they are the tuning techniques that cannot be used as the analysis in advance. SPICE Monte

Carlo (MC) is considered an accurate method for variation analysis, however, it needs extremely large computational effort and is impractical for large designs. Statistical Static Timing Analysis (SSTA) [23–25] was developed about ten years ago and was based on Gaussian gate delay distribution. However, it requires too much run time and memory and has not been widely used in the industry. Then, the On-Chip Variation (OCV) and the Advanced OCV (AOCV) [26] were introduced in the current timing analysis flow and provide sufficient accuracy and risk reduction for Integrated Circuits (IC) designs, but they compare similarly to traditional best-case and worst-case methods, which means they are all corner-based methodology. They can only obtain the path delay variation on the given characterized corners. However, in reality, many un-characterized corners need to be analysed to improve the accuracy of the designs. During such situations, it must take time to characterize the timing library, but the number of signoff corners increases to hundreds or thousands which makes the characterization effort increase too dramatically to withstand. Therefore, it is difficult to trade-off between precision and overhead of the characterizing library. To save characteristic effort and predict path delay variation at multi-corner conditions, many researchers have studied it [16,27]. Alioto et al. [27] regard the delay independent of temperature and voltage at the reference of Fonout of Four (FO4) and it provides a relatively high accuracy and avoids heavy characterization effort. Rithe's et al. [16] method needs less effort to characterize a cell at one corner but cannot predict the timing at a new corner.

Recently, the learning-based method has been widely used in all kinds of fields [28–30], such as optical, image processing and also the Electronics Design Automation (EDA) field, especially timing analysis, and has shown great potential [31–33]. Das et al. [31] build a model that still focuses the cell delay model by a learning-based method that comprehensively captures process, voltage, and temperature, along with input slew and output load, but it is not suited for path delay variation prediction directly. Kahng et al. [32] use a machine learning method to solve the signal integrity (SI) timing problems, which is based on the timing reports from the non-SI mode. It is robust across designs and signoff constraints. Han et al. [33] apply a learning-based method to solve the correlation problem of different timing signoff tools. They develop a learning-based tool to correct the divergence of all kinds of delays at different tools. The applications mentioned all use a learning-based method to solve the timing related issue, which demonstrates the learning method is a promising method to solve the timing prediction problem.

Here, we introduce a novel prediction framework for path delay considering local variation. The framework uses a learning-based method, first to obtain the relationship of the delay variation with circuit features at some corners and, second, to predict new path delay variation at a single corner and unknown corner delay variations at multi-corners. The contributions of the paper are as follows:

- Regarding the single corner, it not only eliminates the characterization effort for the timing library of each cell, but also performs better than AOCV.
- Concerning the multi-corner, the single model setting can be easily expanded to multi-corner, which is not possible in traditional AOCV and MC methods and have less error compared to existing works.

The structure of the paper is as follows: Section 2 introduces the proposed prediction framework for path delay variation based on Machine Learning (ML) from single corner and multi-corner condition perspectives. The experimental setting and results discussion are presented in Section 3. Section 4 concludes this work.

## 2. Proposed Prediction Framework for Path Delay Variation-Based Learning Method

Concerning practical designs, the severe timing constraint is required to satisfy the desired performance. To meet the clock constraint, the prediction of path delay variation is a crucial issue.

A learning-based framework is proposed for the delay variation prediction of a circuit path, without the requirement of corner-based timing libraries, which is illustrated in Figure 1. The first step is data preparation, which mainly contains the generation of train paths. Then, path delay/delay

variation can be obtained by fast SPICE nominal/MC simulation. The basic path features are extracted based on the prepared paths, which are divided into single corner and multi-corner and will be presented in Section 2.2. The last step is the network configuration. The Back Propagation (BP) network [34,35] is selected to build the model due to the problem of regression. The weights and thresholds of every layer and the junction points of the network are acquired through iterations for the train set. The network layers and junction points are adjusted, and the effective features are reselected until the result meets its convergence. The main core steps are training set preparation, features selection and network configuration in the learning-based method.

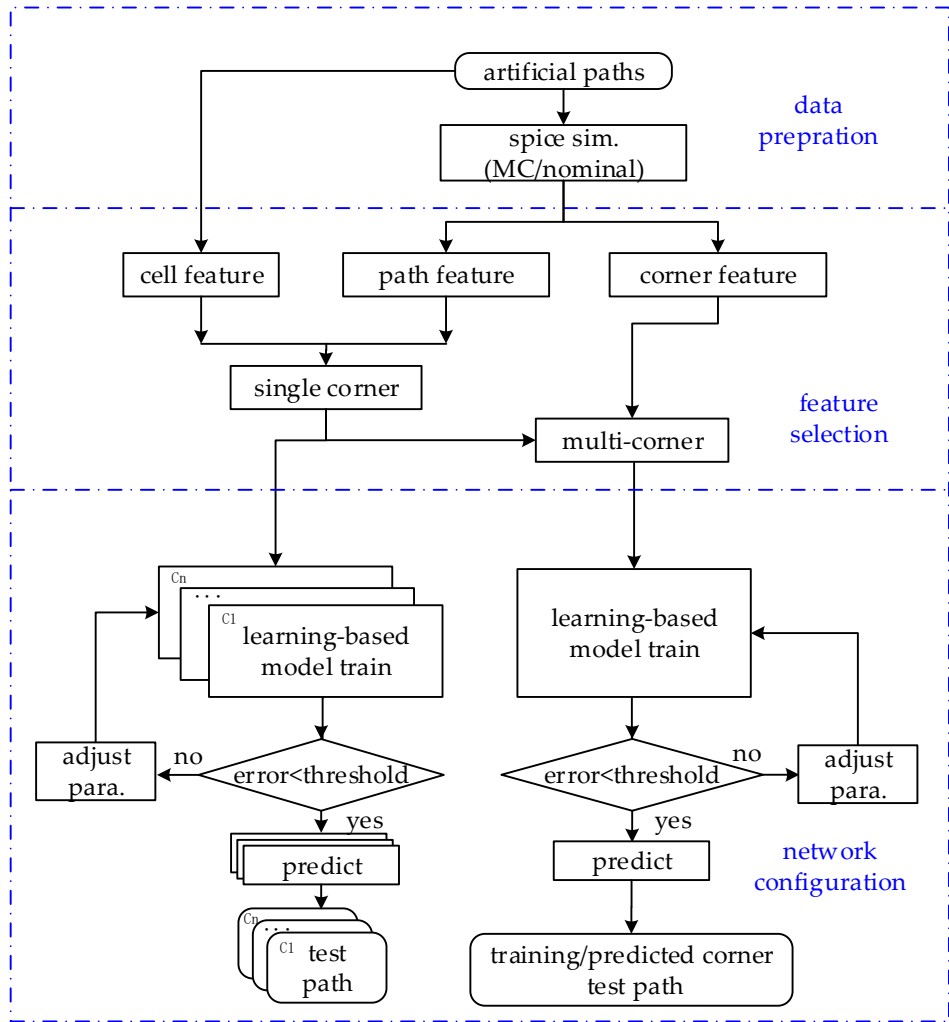

**Figure 1.** The flowchart of the prediction framework for path delay variation based on ML algorithm.

### 2.1. Data Preparation

The first issue faced by the learning-based method is how to construct the training set. Based on the above analysis, the problem we focus on is the path delay variation. To improve the coverage of the training set samples for different circuit parameters and limit the number of MC simulations required for the training set, a group of artificial paths with multiple random gates, sizes, load capacitances, and stages is generated. Meanwhile, the following assumptions are needed during the generation of artificial paths.

(1) All path branches are removed by being replaced with given values for load capacitance.

(2) All values of float input ports of multi-input gates are assumed to be 1 or 0 to ensure it does not influence the signal propagation. Taking the NAND2 gate as an example, one input connects the previous output, and the other input is set to 1.

Under the assumptions mentioned above, all path delay without/with any local variations is measured by SPICE/fast MC simulation based on the generated paths.

## 2.2. Feature Selection

Effective feature selection is another critical step for the learning-based method. The selected features are introduced in the following.

Since the problem of this paper is the regression problem of supervised learning, the input and output features are necessary. The features contain cell topology features, a path delay feature and corner conditions, whose detailed definitions are listed in Table 1. The $td_{x\sigma}$ is the output feature, and the others are input features.

**Table 1.** Feature selection and notation.

| Category | Feature | Notation | Single Corner | Multi-Corner |
|----------|---------|----------|:-------------:|:------------:|
| cell | size | the drive strength of each gate | √ | √ |
| | $N_{stack}$ | the stack transistor number of each gate | √ | √ |
| path | polar | rise of fall of each gate | √ | √ |
| | load | the load capacitance of each stage | √ | √ |
| | td | the nominal delay of each path | √ | √ |
| | $td_{x\sigma}$ | the variation delay of each path at x$\sigma$ | √ | √ |
| corner | T | the temperature of the operation condition | - | √ |
| | V | the voltage of the operation condition | - | √ |

Size and $N_{stack}$ constitute cell topology. Size refers to the width of the transistor, that is, the driving strength of the gate, such as X1, X2 and so on. $N_{stack}$ indicates the stack transistors number of the gate. Random variations across different stacked transistors tend to average out and reduce the variability, compared to a single transistor by the square of $N_{stack}$. Therefore, the $N_{stack}$ is an effective feature of the BP network. A gate has a different $N_{stack}$, however, based on the charging and discharging. Taking NAND2 as an example, the $N_{stack}$ is 2 when discharging, and 1 while charging.

Thus, in a real path, another feature that needs to be introduced is called polar. Polar means representing the transition direction of a gate which is charging or discharging, and it can be obtained according to the propagation characteristics of each gate in the path. The load of each stage and path delay (td) can be easily obtained in the data preparation phase.

Temperature (T) and voltage (V) are the two features that are different from the single-corner and multi-corner conditions. Seen in the single corner prediction the two features are not included, but, in the multi-corner prediction they need to be added.

## 2.3. Network and Configuration

The variation delay of a path can be expressed as follows:

$$td_{x\sigma} = F(size, N_{stack}, polar, load, td, T, V)$$

Concerning a single corner, the feature of $T$ and $V$ can be ignored, but for multi-corners, the $T$ and $V$ are necessary. $F$ is a complex and continuous function. Due to the path delay variation prediction belong to the regression problem, the BP neural network is adopted. The architecture of BP, with three hidden layers, is illustrated in Figure 2, which contains input, hidden and output layers. The input layer concludes the features mentioned in Section 2.2, and the output layer is the path delay variation value. We assume the input layer has $d$ neurons ($x1, \dots, xi, \dots, xd$), the hidden layers has $n$ ($h1_1, h1_2, \dots, h1_j, \dots, h1_n$), $p$ ($h2_1, \dots, h2_k, \dots, h2_p$), $q$ ($h3_1, \dots, h3_t, \dots, h2_q$) neurons respectively, and the output

layers has *l* neurons (*y*). and every active function are all sigmoid, the weights and biases are expressed by *w* and *b*. Additionally, every neuron input and output function is listed in Table 2, where the net and out denote the input and output function of the neuron. It computes the result of every neuron by a forward propagation network and adjusts weights, biases, the number of layers and junction points by back propagation to minimize the total errors of all neurons during the network building.

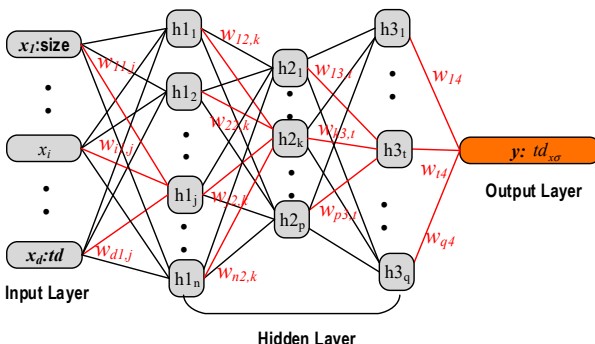

**Figure 2.** The architecture of the BP model.

**Table 2.** Input and Output function of every layer.

| | Function | |
|---|---|---|
| | **Input** | **Output** |
| Hidden 1 layer | $net_{h1_j} = \sum\limits_{i=1}^{d} w_{i1,j} x_i + b_0$ | $out_{h1_j} = \frac{1}{1+e^{-net_{h1_j}}}$ |
| Hidden 2 layer | $net_{h2_k} = \sum\limits_{j=1}^{n} w_{j2,k} out_{h1_j} + b_1$ | $out_{h2_k} = \frac{1}{1+e^{-net_{h2_k}}}$ |
| Hidden 3 layer | $net_{h3_t} = \sum\limits_{k=1}^{p} w_{k3,t} out_{h2_k} + b_2$ | $out_{h3_t} = \frac{1}{1+e^{-net_{h3_t}}}$ |
| Output layer | $net_y = \sum\limits_{t=1}^{q} w_{t3} out_{h3_t} + b_3$ | $out_y = \frac{1}{1+e^{-net_y}}$ |

## 3. Experimental Results and Discussions

Experiments were carried out under the process of Semiconductor Manufactory International Corporation (SMIC) 40 nm. Considering training and testing, plenty of paths were generated with randomly chosen cells and connecting structure, as described in Section 2. The path delay variation was characterized by 1% and 99% quantile with MC simulation for 1000 times. Regarding the training set, we chose a wide range of temperatures (0–25 °C) and voltages (0.6–1.1 V) to provide a reliable reference boundary which met the need of practical use for the model. The results of the predicting ability of the model are described through the mean relative error of the testing set, which is defined as follows:

$$error = \frac{1}{n} \sum_{1}^{n} \left| \frac{y_i - \widehat{y}_i}{y_i} \right|$$

where $y_i$ is the actual path delay variation value, and $\widehat{y}_i$ is the predicted value.

### 3.1. Path Delay Variation Prediction at a Single Corner

Path delay variation prediction at a single corner means that different models are built at each corresponding corner. Taking 5 stage path as an example, a detailed analysis of the training sample number and path delay accuracy are described in the following.

### 3.1.1. The Selection of Sample Number

It is commonly known that the more training data the network has, the higher the accuracy it will get. Enormous training data will have a more prominent cost, however. Looking at the training set, $td_{-x\sigma}$ and $td_{+x\sigma}$, every path is acquired by executing 1000 times MC simulations, on the one hand,, while, on the other hand, a larger training set will need more iterations to reach convergence for the network, which means more training time. Therefore, the trade-off between training sampling numbers and accuracy should be considered.

To attain the appropriate sample numbers for training, the analysis of the testing error curve with the number of training samples was adopted. To ensure the training sample number for our network, a large scope of training sample numbers was chosen, from 10 to 2500 with a step of 10. Figure 3 shows the x-axis indicates the training number, and the average error of the testing set (black line) and training set (red line) at this training sample is shown in the y-axis. Moreover, the curves are portrayed in several corners. We observed that for different corners, after 800 training samples, the testing error would decrease very slowly and this value was very close within different corners. Thus, we chose 700 as the number of training samples and 100 as the validation samples for saving SPICE simulation time and network training time in the context of guaranteed precision.

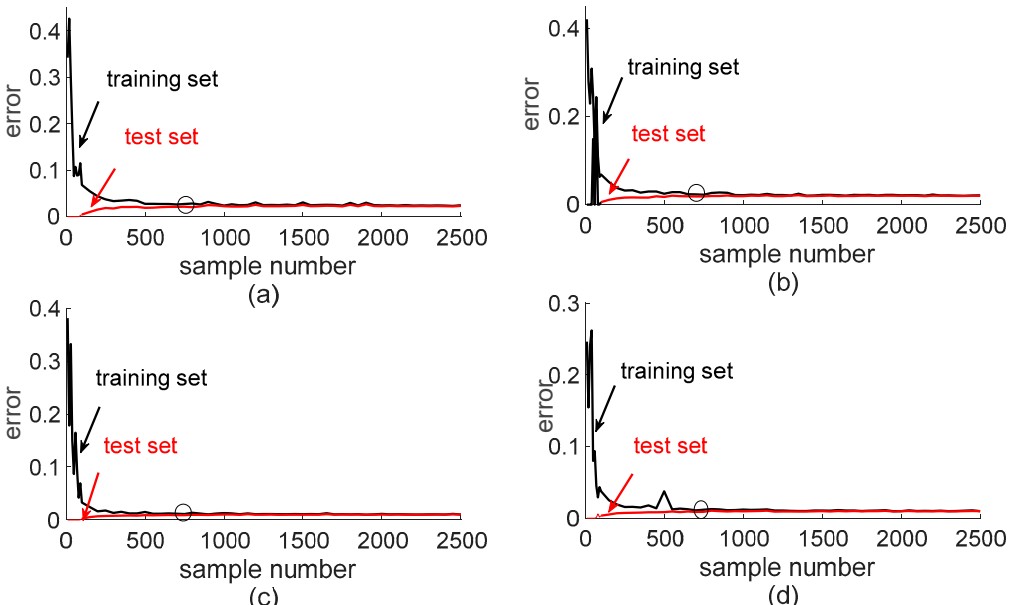

**Figure 3.** The trade-off of sample number and accuracy at different conditions. (**a**) 0.6 V 25 °C, (**b**) 0.6 V 75 °C, (**c**) 0.8 V 25 °C, (**d**) 0.8 V 75 °C.

### 3.1.2. Path Delay Variation Prediction at Single Corner

The errors of the path variation delay across the two methods are illustrated in Figures 4 and 5. One is the traditional method (AOCV), which predicts the worst case of the path delay variation by summarizing the nominal delays of all cells in the path with a derating coefficient for each stage. The other is the proposed method (BP-based), which predicts the delay of a new path by a network established by the training set. The x-axis represents the different operation conditions combined with all kinds of voltage and temperature, and the y-axis represents the mean error of the different methods.

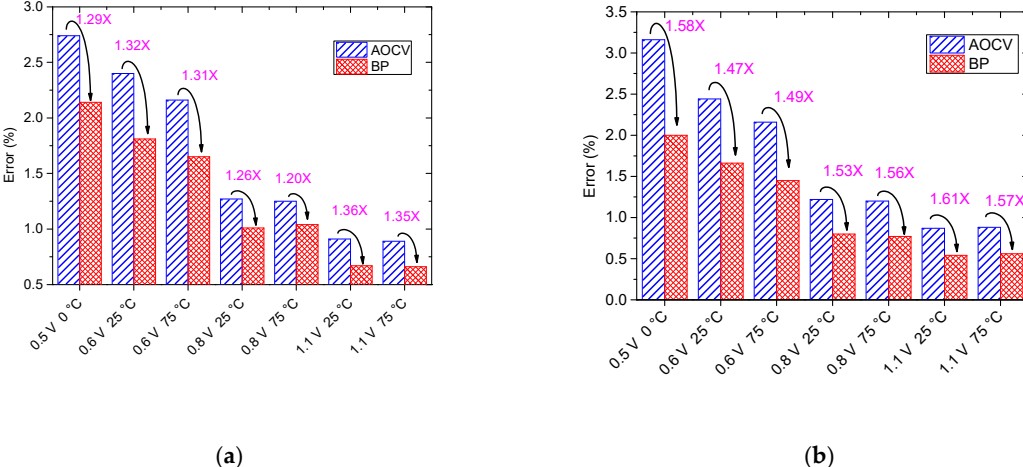

**Figure 4.** Path variation delay training error at $td_{-x\sigma}$ at a single corner. (**a**) train error; (**b**) test error.

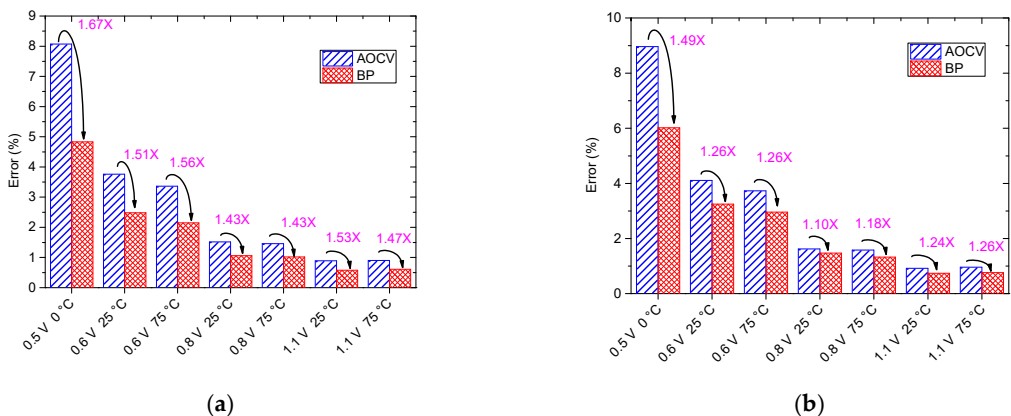

**Figure 5.** Path variation delay predicted error at $td_{+x\sigma}$ at a single corner. (**a**) train error; (**b**)test error.

Viewing the Figures, we can see that no matter if the probability is $td_{-x\sigma}$ or $td_{+x\sigma}$, and no matter if the error is the train set or test set, the BP model performs better than AOCV. Regarding $td_{-x\sigma}$, the error in BP decreases by 1.20X–1.36X less than AOCV for the train set, and by 1.47X–1.58X for the test set. Concerning $td_{+x\sigma}$, the error in BP decreases by 1.43X–1.67X less than AOCV for the train set, and by 1.10X–1.49X for the test set. Therefore, it decreases by 1.4X on average. Besides, as the voltage decreases, the mean errors of the two methods are increasing, which is due to the non-gaussian distribution at low voltage. However, compared with AOCV, the error of the BP model increases more slowly, due to the network build of the BP model containing a complex non-linear operation base on the non-linear active function, which further demonstrates that BP is a more suitable method for the path delay variation prediction at low voltage.

### 3.2. Path Delay Variation Prediction at Multi-Corner

A single corner demonstrates that the learning-based method performs better than the AOCV method. It should be noted that the AOCV cannot be used in multi-corners to predict unknown corners, so only the result of the BP method is illustrated here. Furthermore, the framework is extended to the multi-corner condition and to predict the corner which is not included in the training set. Table 3 lists the training corners and predicted corners, where ▲ indicates training corners and ★ expresses predicted corners. Also, the training corners contain train paths and test paths that are characterized in advance, and the predicted corners include new corners and new paths that are different from those in the training set to show the advantage of generalization in the proposed work.

**Table 3.** Training and predicted corners.

| T (°C) \ V | 0.6 | 0.7 | 0.8 | 0.9 | 1.0 | 1.1 |
|---|---|---|---|---|---|---|
| 0 | ▲ | | ▲ | | ★ | ▲ |
| 20 | | ★ | | | | |
| 25 | ▲ | | ▲ | | | ▲ |
| 50 | ★ | ★ | | ★ | | |
| 75 | ▲ | | ▲ | | | ▲ |
| 100 | | ★ | | | ★ | |
| 125 | ▲ | | ▲ | | | ▲ |

▲: training corner ★: predicted corner.

The two kinds of stages (5 and 10) are evaluated, and the mean error of the training corners and the predicted corners are shown in Figure 6. The errors of $td_{-x\sigma}$ and $td_{+x\sigma}$ increase with decreasing voltage and temperature at the two kinds of stages. Regarding $td_{-x\sigma}$, the error increases from 1.21% to 2.82% for stage 5 and from 0.63% to 1.32% for stage 10. Concerning $td_{+x\sigma}$, the error increases from 1.35% to 4.59% for stage 5 and from 0.6% to 1.82% for stage 10. Besides, the errors in stage 10 are less than stage 5 because, as the number of stages increases, the non-Gaussian phenomenon gradually becomes smaller and the interaction between each stage plays a complementary role, so the error itself will be reduced.

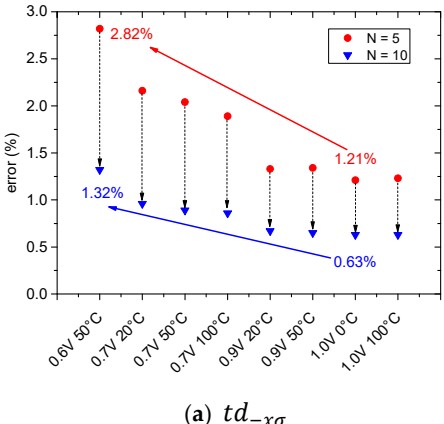

(a) $td_{-x\sigma}$

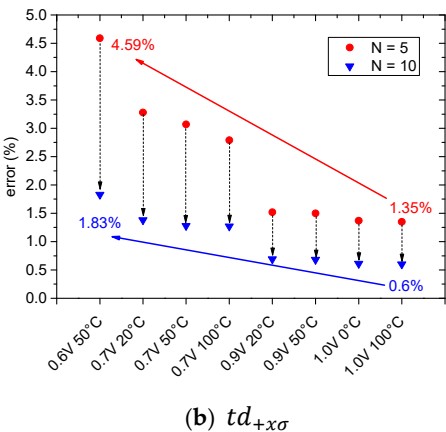

(b) $td_{+x\sigma}$

**Figure 6.** Path variation delay mean error prediction of stage 5 and 10 at multi-corner conditions. (a) $td_{-x\sigma}$; (b) $td_{+x\sigma}$.

Additionally, the errors are less than those found in previous works, and the detailed comparison is introduced in the following. The proposed prediction method is compared with some different ways proposed in other papers. Table 4 shows the comparison of our work with that by Alioto et al. [27] and Rithe's et al. [16]. All three works have good accuracy for the prediction of a given path at a given temperature (T) and voltage (V), especially in near-threshold voltage, of which the distribution is non-Gaussian, and their average prediction errors are all below 6%. The common contributions of the three works are that they all provide a way to predict path delay variation and avoid great timing cost of the MC method.

**Table 4.** Comparison with previous works.

|  | [16] | [25] | This Work |
|---|---|---|---|
| Technology | 28 nm | 28, 40, 65 nm | 40 nm |
| V | 0.5 V | 0.3–1.2 V | 0.6–1.1 V |
| T | N/A | 70 °C | 0–125 °C |
| Error | 5.68% (6.21%) | 5.3% (11.4%) | 2% (4.59%) |
| Characterize effort | 100SPICE/cell | 2000SPICE/cell | 1000SPICE/path |
| Application range | Single T,V | Multiple T,V | Multiple T,V |

Compared to the work by Rithe's et al. [16], the main improvement of our method is that it can handle any temperatures and voltages, as long as the model is established, while in the work by Rithe's et al. [16], the method is based on the gate timing library and rebuilding all the gate characteristic of a path at a new temperature or voltage. Therefore, the other method is not applicable for multi-corners. Compared to the study by Alioto et al. [27], which regards the delay independent from temperature and voltage at the reference of Fanout of 4 (FO4), it has a good result at multiple temperatures and voltages. Regarding our work, which employed ML, it models the relationships between temperature, voltage and path delay, so it gains a higher accuracy of 2%, on average.

When considering the timing cost for path delay variation prediction, the study by Rithe's et al. [16] characterizes the statistical delay for every cell in one corner, which will spend about 100 times of SPICE simulations, it can predict any new path delay variation at this corner with the average(max) error of 5.68% (6.21%). When the path delay variation of other corners is needed, the number of SPICE simulations will multiply the number of corners. The FO4 method proposed by Alioto et al. [27] needs 2000 SPICE simulations for each gate characteristic and is appropriate for multiple temperatures and voltages, that is to say, after characterizing every cell, the method can predict any new path at multiple corners within the average(max) error of 5.3% (11.4%). The timing cost of our method mainly stays in the training set build, which needs 1000 SPICE simulations for each path in the training set. Subsequent to the training, the model can provide a new path delay in a new corner with an average(max) of 2% (4.59%), which is the lowest/minimum average error of these methods. Thus, our work provides a promising method for solving the path delay variation problem for single- and multi-corner conditions.

## 4. Conclusions

Here, the learning-based method was adopted to solve the path delay variation problem under single and multiple corners. First, the characterization effort for the timing library of each cell was eliminated in our work, as well as the computing effort of the non-grid timing by interpolation. Second, the proposed framework predicted the path delay variation at single- and multi-corners. Regarding the single corner, the train/test error was much lower than AOCV, while for multi-corner, the error in stage 10 was much less than that in stage 5, but the predicted errors were within the acceptable range compared with previous works.

**Author Contributions:** J.G., P.C. and J.Y. organized this work. J.G., Z.S., B.X. and Z.L. performed the modeling, simulation and experiment work. The manuscript was written by J.G. and P.C., and edited by J.G. All authors have read and agreed to the published version of the manuscript.

**Funding:** This work was supported in part by the National Key Research and Development Program of China (Grant No. 2019YFB2205004), and National Natural Science Foundation of China (Grant No. 61834002), and the National Science and Technology Major Project (Grant No. 2017ZX01030101).

**Acknowledgments:** The authors thank Jiangping Wu and Hao Yan for their helpful insight and suggestions.

**Conflicts of Interest:** The authors declare no conflict of interest.

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
