# Peer review of "Novel Prediction Framework for Path Delay Variation Based on Learning Method"

_electronics, doi:10.3390/electronics9010157_

Round 1

Reviewer 1 Report

1. The author manuscript said “The variation delay of a path can be expressed as follows:. tdxσ=F(size, Nstack, plar, load, td, T, V)”

Please write down the F(size, Nstack, plar, load, td, T, V).

2. The author manuscript said Figure 2. The architecture of BP model.

Please show the input of the BP model and function of input, hidden and output layers.

3. The author manuscript said “Figure 3. The tradeoff of sample number and accuracy.”

Please write the function.

4. The learning method was already the BP modal.

    This is not good to publish the electronics.

5. The given BP model is too easy to set.

    This is not good to publish the electronics.

Reviewer 2 Report

The manuscript proposes a framework that can predict the path delay variation at a single and some unknown corners, using a learning-based method in order to obtain the relationship of the delay variation with circuit features at a corner.

It is well orginized, it describes the architecture as well as the proposed prediction method is compared with some different ways proposed in other papers.

In my point of view the paper is suitable for publication

Reviewer 3 Report

In this paper, the authors introduced a novel prediction framework for path delay considering local variation is introduced. The framework uses a learning-based method to obtain the relationship of the delay variation with circuit features at some corners and to predict new path delay variation at a single corner and unknown corner delay variation at multi-corner.

It shows some interesting results. The work is not acceptable in its present form, however, before the publication, there are few questions and suggestions.  The final decision depends on the authors' response.

1.The information for the flowchart of the prediction framework for path delay variation based on ML algorithm is not enough. The authors should give much more information about this. So the readers can get its reproducibility. 

2. The authors should give much more information about the novelty of this paper, especially the effect of using this prediction framework for path delay variation, why not use a classical Monta Calero?

3. More references need to be included in the introduction part to understand the applications of this topic.

"Neural networks within multi-core optic fibers", Scientific Reports 6, 2016 (article no.29080). "Improved diagnostic process of multiple sclerosis using automated detection and selected process in Magnetic Resonance Imaging",Applied Sciences, Issue 7(8), 2017, 831 (13 pages). "Improving Raman spectra of pure silicon using super-resolved method", J. of Optics, 21(7), 2019 (075801 – 6 pages)

4. Much more discussion about the results should be given in this paper, especially the author needs to provide enough physicals mechanism analysis about the results.

Round 2

Reviewer 3 Report

The new version can be published only need to modify Ref 28, there is a space in the word optic